# The Emerging Nosocomial Pathogen *Klebsiella michiganensis*: Genetic Analysis of a KPC-3 Producing Strain Isolated from Venus Clam

Serena Simoni,[a] Francesca Leoni,[b] Laura Veschetti,[c] Giovanni Malerba,[c] Maria Carelli,[d] Maria M. Lleò,[d] ⓘ Andrea Brenciani,[e] ⓘ Gianluca Morroni,[e] Eleonora Giovanetti,[a] Elena Rocchegiani,[b] Francesca Barchiesi,[b] ⓘ Carla Vignaroli[a]

[a]Department of Life and Environmental Sciences, Polytechnic University of Marche, Ancona, Italy

[b]Istituto Zooprofilattico Sperimentale dell'Umbria e delle Marche, Sezione di Ancona, Laboratorio Nazionale di Riferimento (LNR) per il Controllo delle Contaminazioni Batteriche dei Molluschi Bivalvi Vivi, Ancona, Italy

[c]Department of Neurosciences, Biomedicine and Movement Sciences, University of Verona, Verona, Italy

[d]Department of Diagnostics and Public Health, University of Verona, Verona, Italy

[e]Department of Biomedical Sciences and Public Health, Polytechnic University of Marche, Ancona, Italy

**ABSTRACT** The recovery and characterization of a multidrug-resistant, KPC-3-producing *Klebsiella michiganensis* that was obtained from Venus clam samples is reported in this study. A whole-genome sequencing (WGS) analysis using Illumina and Nanopore technologies of the *K. michiganensis* 23999A2 isolate revealed that the strain belonged to the new sequence type 382 (ST382) and carried seven plasmid replicon sequences, including four IncF type plasmids (FII, FIIY, FIIk, and FIB), one IncHI1 plasmid, and two Col plasmids. The FIB and FII$_k$ plasmids showed high homology to each other and to multireplicon pKpQIL-like plasmids that are found in epidemic KPC-*K. pneumoniae* clones worldwide. The strain carried multiple $\beta$-lactamase genes on the IncF plasmids: $bla_{OXA-9}$ and $bla_{TEM-1A}$ on FIB, $bla_{KPC-3}$ inserted in a Tn4401a on FII$_K$, and $bla_{SHV-12}$ on FII$_Y$. The IncHI1-ST11 harbored no resistance gene. The curing of the strain caused the loss of all of the *bla* genes and a rearrangement of the IncF plasmids. Conjugal transfer of the $bla_{OXA-9}$, $bla_{TEM-1A}$ and $bla_{KPC-3}$ genes occurred at a frequency of $5 \times 10^{-7}$, using *K. quasipneumoniae* as a recipient, and all of the *bla* genes were transferred through a pKpQIL that originated from the recombination of the FIB and FIIk plasmids of the donor. A comparison with 31 *K. michiganensis* genomes that are available in the NCBI database showed that the closest phylogenetic relatives of *K. michiganensis* 23999A2 are an environmental isolate from soil in South Korea and a clinical isolate from human sputum in Japan. Finally, a pan-genome analysis showed a large accessory genome of the strain as well as the great genomic plasticity of the *K. michiganensis* species.

**IMPORTANCE** *Klebsiella michiganensis* is an emerging nosocomial pathogen, and, so far, few studies describe isolates of clinical origin in the environment. This study contributes to the understanding of how the dissemination of carbapenem-resistance outside the hospital setting may be related to the circulation of pKpQIL-like plasmids that are derived from epidemic *Klebsiella pneumoniae* strains. The recovery of a carbapenem-resistant isolate in clams is of great concern, as bivalves could represent vehicles of transmission of pathogens and resistance genes to humans via the food chain. The study demonstrates the plasticity of *K. michiganensis* genome, which is probably useful to multiple environment adaptation and to the evolution of the species.

**KEYWORDS** *Klebsiella michiganensis*, carbapenem-resistance, KPC-3, pKpQIL plasmids

Address correspondence to Carla Vignaroli, c.vignaroli@univpm.it.

The authors declare no conflict of interest.

The genus *Klebsiella* includes well-known bacterial species that are common causes of hospital acquired infections as well as species that are largely distributed in numerous environmental habitats. *Klebsiella pneumoniae* is one of the most important nosocomial

pathogens contributing to the increasing prevalence of multidrug-resistant strains (1). Carbapenem resistance in *K. pneumoniae* is of particular concern, as carbapenems are last-line antibiotics for the treatment of serious human infections (2). Carbapenemase-producing *Enterobacteriaceae* (CPE) are being increasingly reported, and the dissemination of the different enzymes (mainly KPC, VIM, OXA, and NDM) has been mostly associated with the spread of *K. pneumoniae* epidemic clones that carry *bla* genes on mobile elements and conjugative plasmids (3). The number and diversity of plasmids in *K. pneumoniae* strains is typically greater than those in other Gram-negative bacteria, suggesting its plasmid permissive traits (1). Acquired resistance genes are frequently carried by large conjugative plasmids that belong to a limited number of plasmids from the Inc group. In particular, FII, N, R, X3, L/M, and A/C replicons have been reported as the prevalent plasmid groups carrying carbapenemase and Extended-Spectrum Beta-Lactamase (ESBL) genes (4). Moreover, multireplicon and mosaic plasmids, such as pKpQIL and pKPN3, respectively, have been described in epidemic *K. pneumoniae* strains that have contributed to KPC-type dissemination (4).

Recently, *Klebsiella michiganensis*, which is closely related to the *Klebsiella oxytoca* species, has been reported as an emerging opportunistic, nosocomial pathogen, and clinical isolates carrying carbapenemase-encoding genes have been described (5–8). Environmental reservoirs of *K. michiganensis* have been poorly investigated, even if they are largely found in many natural habitats, as are other *Klebsiella* species. *K. michiganensis* isolates have also been recovered from wastewaters (9, 10), suggesting that coastal areas receiving urban and hospital effluents could represent potential sources of infections for humans. In central Italy, along the coast of the Adriatic Sea, *Escherichia coli* epidemic clones of human origin have been isolated from waters, sediments, and mollusks (11–13). Bivalves have the potential to concentrate microorganisms as a consequence of their filter-feeding activity. Therefore, when they grow in polluted coastal areas, they may represent secondary reservoirs of pathogens, such as carbapenem-resistant *Enterobacteriaceae* (14). Many bivalve species are consumed raw or lightly cooked, so they could represent vehicles of transmission of bacteria and resistance genes to humans via the food chain. Moreover, to date, there is no European Union (EU) monitoring of antibiotic resistance in bacteria isolated from bivalves and seafood, and authorities, such as the European Food Safety Authority (EFSA), underpin the urgent need to reduce and eliminate resistant bacteria in all food production sectors (15).

In this study, a KPC-producing strain of *K. michiganensis* isolated from clams was characterized via whole-genome sequencing (WGS). Clams were collected from natural beds along the coast of the Adriatic Sea in central Italy. Genetic elements carrying $\beta$-lactam-resistance genes were analyzed, and the ability of the strain to transfer carbapenem resistance via conjugation was also investigated. Finally, we performed a comparison with publicly available *K. michiganensis* complete genome sequences to investigate the relatedness of our isolate to environmental and clinical *K. michiganensis* strains.

## RESULTS AND DISCUSSION

**Identification of a carbapenem-resistant isolate.** From the screening of clam samples for the detection of CPE, a single isolate, designated 23999A2 was selected. This isolate was able to grow in agar supplemented with ertapenem and was positive to the modified Hodge test. The determination of the minimum inhibitory concentration (MIC) values for ertapenem, imipenem, and meropenem showed that the isolate was resistant to the three carbapenems with MIC values of 256, 16, and 8 $\mu$g/mL, respectively. Carbapenem resistance was also confirmed via polymerase chain reaction (PCR) for the presence of a $bla_{KPC}$ gene encoding a KPC-type carbapenemase. The unusual isolation of a CPE from bivalves was the input to proceed with the full characterization of the strain via WGS. The isolate was initially identified by MALDI TOF-mass spectrometry as a *Klebsiella oxytoca*, but after genome sequencing, an OrthoANI analysis assigned the strain to the *K. michiganensis* species, based on its 98.82% identity to the reference *K. michiganensis* KCTC1686 strain (accession no. CP003218). The membership of the strain 23999A2 to the *K. michiganensis* species was also confirmed by uploading the genome data to the rMLST website. The assembled

**TABLE 1** Main characteristics of the plasmids carried by the *K. michiganensis* and *K. quasipneumoniae* strains analyzed in this study

| Plasmid | Inc group | Length (bp) | AR genes | Reference |
|---|---|---|---|---|
| *K. michiganensis* 23999A2 (Donor strain) | | | | |
| pKm_4 | HI1 | 328,801 | $-^a$ | This study |
| pKm_5 | FII$_Y$ | 151,147 | $bla_{SHV-12}$, *mer* operon | This study |
| pKm_6 | FII$_K$ | 140,416 | $bla_{KPC-3}$ | This study |
| pKm_7 | FII | 69,358 | - | This study |
| pKm_8 | FIB | 49,652 | $bla_{OXA-9}$, $bla_{TEM-1A}$, Δ*mer* operon | This study |
| *K. michiganensis* 23999A2 cured strain | | | | |
| pKm_4 | HI1 | 328,804 | - | This study |
| pKm_5b | FII$_Y$ | 123,928 | - | This study |
| pKm_8b | FIB | 78,517 | - | This study |
| pKm_7 | FII | 69,360 | - | This study |
| *K. quasipneumoniae* ATCC700603-Col$^R$ (recipient strain) | | | | |
| pKQPS1 | FII$_K$ | 152,001 | - | 20 |
| pKQPS2 | M | 77,127 | $bla_{OXA-2}$, $bla_{SHV-18}$ | 20 |
| pMCR-1.2-IT-Ec | X4 | 33,293 | *mcr-1.2* | 19 |
| *K. quasipneumoniae* tc-1 (transconjugant) | | | | |
| pKQPS1 | FII$_K$ | 151,998 | - | This study |
| pKQPS2 | M | 77,110 | $bla_{OXA-2}$, $bla_{SHV-18}$ | This study |
| pMCR-1.2-IT-Ec | X4 | 33,303 | *mcr-1.2* | This study |
| pKQTC-1 | FIB-FII$_k$ | 113,639 | $bla_{OXA-9}$, $bla_{TEM-1A}$, $bla_{KPC-3}$, Δ*mer* operon | This study |

$^a$Dash means no resistance genes detected.

genome was also submitted to the PubMLST of the *K. oxytoca* species complex for the determination of its sequence type (ST). The *K. michiganensis* 23999A2 isolate showed two new alleles (phoE-107 and tonB-89), and it was deposited into the database for its assignment to a new ST (ST382).

**WGS results.** The results obtained from both Illumina and Nanopore sequencing technology evidenced a genome consisting of 6,619,970 bp with a 58.85% guanine-cytosine (GC) content. At least seven plasmid replicon sequences were found by PlasmidFinder, including four IncF group (FII, FII$_Y$, FII$_K$, and FIB), one IncHI1, and two Col (Col440II and Col [pHAD28]) plasmids.

The analysis of the WGS data using ResFinder indicated that the *K. michiganensis* strain carried four β-lactamase genes: $bla_{OXA-9}$, $bla_{TEM-1A}$, $bla_{SHV-12}$, and $bla_{KPC-3}$, all of which were on plasmid contigs and two other resistance genes that mediate fluoroquinolone (*oqxB*) and fosfomycin (*fosA*) resistance and are located on the chromosomal 3,055,291 bp contig. The major characteristics of the *K. michiganensis* plasmids are reported in Table 1.

The $bla_{OXA-9}$ and $bla_{TEM-1A}$ genes were colocalized on the 49,652-bp pKm_8 plasmid that contained the FIB replicon, whereas the $bla_{KPC-3}$ gene was typically inserted on the transposon Tn*4401a*, which was located on a large plasmid (pKm_6) of 140,416 bp that contained the FII$_K$ replicon. The pKm_8 was identical to corresponding regions of many pKpQIL-like plasmids, described as multireplicon IncFIB-IncFII$_K$ in *K. pneumoniae* epidemic clones, such as ST258 or ST307 (1, 16). The pKm_6 also exhibited the highest coverage (88%) and identity (99,6%) with a pKpQIL plasmid (accession no. CP014765.1) found in a *K. pneumoniae* human strain. As shown in Fig. 1, *bla* gene contexts on pKm_6 and pKm_8 showed 100% sequence identity with the corresponding regions of the pKpQIL plasmid (pKPC) of the *K. pneumoniae* D1 strain (accession no. CP043971.1). In addition, pKm_6 and pKm_8 shared more than 19,000 bp with >95% identity. The high similarity of pKm_6 and pKm_8 to each other and to pKpQIL-like plasmids suggests that they could both be derivatives of a pKpQIL replicon. These elements could be the result of recombination events between homologous regions on different IncF plasmids that are carried by the same cell, as reported in literature (4). The $bla_{SHV-12}$ gene was carried by a plasmid of 151,147 bp (pKm_5) that

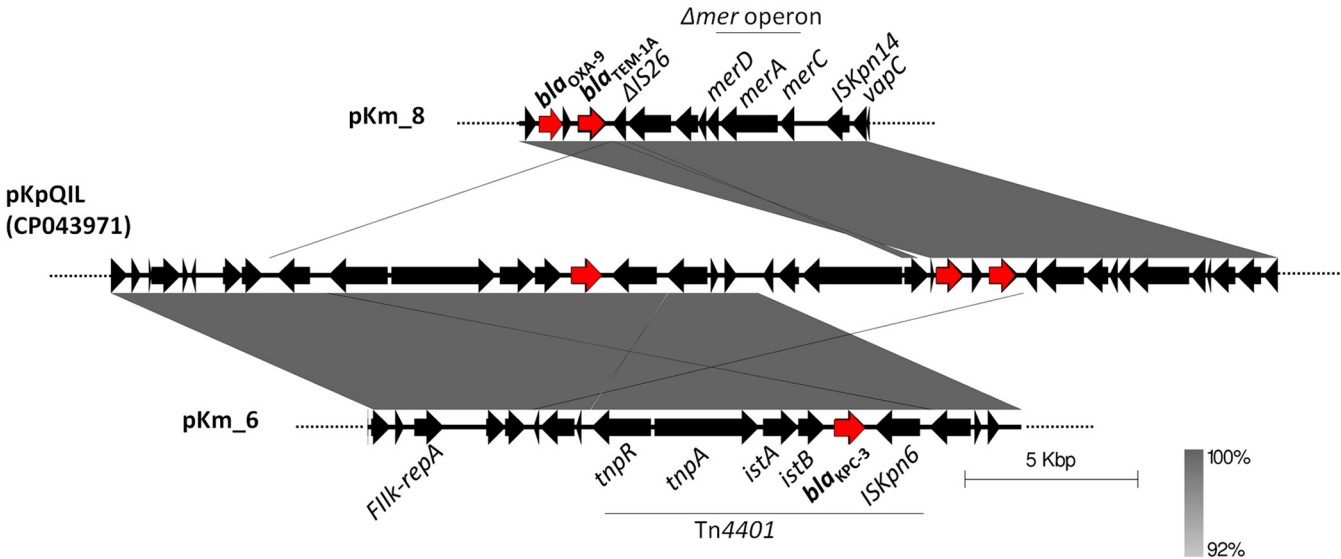

**FIG 1** Alignment of the $bla_{KPC-3}$ and $bla_{OXA-9}$-$bla_{TEM-1A}$ contexts of pKm_6 and pKm_8, respectively, with the corresponding regions of a multireplicon pKpQIL plasmid of *K. pneumoniae* (GenBank accession no. CP043971). The gray shading indicates regions of shared homology (ranging from 92 to 96% to 100%). Red arrows indicate the *bla* resistance genes.

contained the FII$_Y$ sequence of the IncF family. This plasmid shared 84% coverage and 99.96% nucleotide identity with the plasmid pRHBSTW-00909_3 (accession no. CP056138.1), of 166,165 bp in size, that was recovered from an animal *K. michiganensis* strain (17). The $bla_{SHV-12}$, together with a recombinase (*hin*) and a transposase (*tnpA*) encoding gene, were comprised in an approximately 7,300-bp region that was delimited by two IS26 with opposite orientations. A complete *mer* operon conferring mercury resistance was observed at approximately 1,500 bp upstream of the composite transposon carrying the $bla_{SHV-12}$ gene. The *mer* operon was bracketed by two further IS, represented by the IS4321R (upstream) and IS26 (downstream) (Fig. 2). Interestingly, on the pKm_8 plasmid, a truncated copy of the *mer* operon downstream of $bla_{TEM-1A}$ was found; this arrangement is a common trait of pKpQIL plasmids (4). Also, the truncated operon was delimited by two IS elements: a deleted IS26 on the left and an ISKpn14 on the right (Fig. 1). These findings also suggested that the pKm_8 plasmid could be the result of genetic recombination events among IncF plasmids that were harbored by a prior cell.

The fourth IncF plasmid (pKm_7), of 69,358 bp in size, contained the FII replicon and carried no resistance genes. It was highly similar (93% coverage and 97% identity) to a plasmid (accession no. CP058121.1) described in a *K. michiganensis* strain of animal origin (17).

The IncHI1 plasmid (pKm_4), of 328,801 bp in size, was typed on the Plasmid PubMLST database (https://pubmlst.org/) as an IncHI1-ST11 and showed the highest coverage (91%) and nucleotide identity (99.9%) with a plasmid (accession no. CP056496.1) identified in an animal strain of *K. oxytoca* (17). The plasmid carried a gene encoding a tellurium ion resistance protein (*terC*) but no antimicrobial resistance genes.

**Curing and conjugation experiments.** After 37 passages on ertapenem-free agar plates, the strain was cured of its 140-kb plasmid (pKm_6) that was carrying the $bla_{KPC-3}$ gene. The loss of $bla_{KPC-3}$ was demonstrated by antibiotic susceptibility to carbapenem and via WGS.

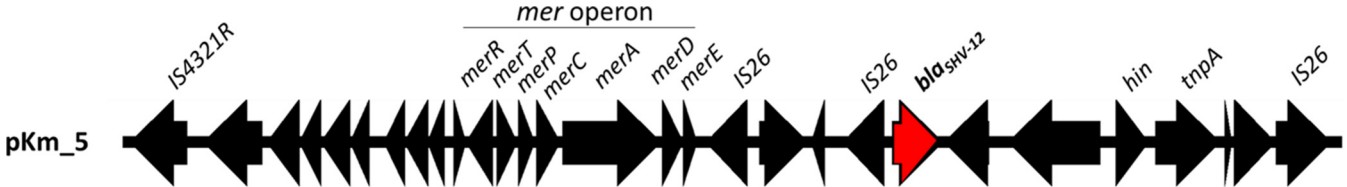

**FIG 2** Genetic context of the *mer* operon and $bla_{SHV-12}$ gene (highlighted in red) in the pKm_5 of *K. michiganensis* 23999A2.

A PlasmidFinder analysis showed that in the cured strain, the $FII_K$ replicon sequence was lost, too, in contrast to FIB, $FII_Y$, FII, and HI1, which were still present. The strain still harbored the IncHI1 plasmid of 328 kb and the IncFII plasmid of 69 kb that were carrying no resistance genes, whereas all of the other IncF plasmids were rearranged in replicons of different sizes from those of the wild-type strain (Table 1). Interestingly all of the *bla* genes and the *mer* operon were lost during the numerous passages in the antibiotic-free agar plates.

The transferability of carbapenem resistance was investigated via mating experiments using *E. coli* 1816 or a derivative strain of the *K. quasipneumoniae* ATCC700603. The *K. michiganensis* strain was able to transfer carbapenem resistance only to a *K. quasipneumoniae* recipient with a frequency of $5 \times 10^{-7}$. The transconjugants were positive to $bla_{KPC-3}$, $bla_{TEM-1}$, and $bla_{OXA-9}$. Also, the FIB and $FII_K$ replicons carried by the pKm_6 and pKm_8 plasmids of the donor were present in the transconjugants. A WGS analysis of the transconjugant *K. quasipneumoniae* tc-1 showed that the isolate retained the three resident plasmids and acquired a single plasmid (named pKQTC-1) of 113,639 kb that was carrying both the FIB and $FII_K$ replicon genes as well as all three of the *bla* genes ($bla_{KPC-3}$, $bla_{TEM-1}$, and $bla_{OXA-9}$) (Table 1). The pKQTC-1 plasmid showed partial homology with all four of the IncF plasmids of the donor but, in particular, shared 74% and 44% coverage with pKm_6 and pKm_8, respectively, with a 99.9 to 100% sequence identity (Fig. 3). A recombination event among the pKm_6 and pKm_8 of the donor, during or prior to conjugation, could have occurred and led to the formation of the pKQTC-1, a multiresistant and multireplicon pKpQIL-like plasmid. Indeed, a BLASTn analysis confirmed the high homology of pKQTC-1 to many of the pKpQIL of *K. pneumoniae* as well as the 100% coverage and sequence identity to pKPC (accession no. CP043971.1).

Rearrangements are often IS-mediated, and, in particular, IS*26* could have a critical role in the steady reorganization of new pKpQIL variants, as reported elsewhere (18, 19).

**Comparative genomic analysis.** A comparative genomic analysis was also performed to study the genomic relatedness with the clinical and environmental *K. michiganensis* complete genomes that are available on NCBI (*n* = 31; see Table S1 for details). Interestingly, 4 strains showed an ANI of <95% against all of the analyzed genomes, suggesting that they likely belong to a different species. Thus, they have been excluded from the following analyses. Among the 27 *K. michiganensis* genomes having an ANI of ≥95%, 16 were clinical strains, 9 were environmental strains, and 2 strains had no metadata regarding the NCBI "Sample relevance" field.

A phylogenetic analysis (interactive phylogenetic tree available at https://microreact.org/project/nyYxibVGLjgkN2hzx8WfBd/19371e9e) showed that the genomes did not cluster based on either isolation source or geographical area. In particular, the closest genome sequences to *K. michiganensis* 23999A2 were those collected from an environmental isolate that was sampled from soil in South Korea and from a clinical isolate that was sampled from human sputum in Japan (respective accession numbers: CP041515.1 and AP022547.1).

In order to further investigate possible genomic similarities among *K. michiganensis* 23999A2 and clinical or environmental isolates in terms of resistome, virulome, and mobilome, we assessed the presence of antimicrobial resistance genes, virulence genes, plasmids, and phages (Fig. 4). Once again, no clear clustering was observable between the clinical and environmental isolate sequences or among the different geographic areas. Interestingly, *K. michiganensis* 23999A2 did not carry any $bla_{OXY}$ genes, which were present in all of the analyzed genomes and has also been reported in the closely related *K. oxytoca* species (20). Conversely, two additional chromosomal genes encoding $\beta$-lactamase family proteins were found on *K. michiganensis* 23999A2 and were highly conserved in many *K. michiganensis* and *K. oxytoca* strains (99 to 100% sequence identity), as was shown via a BLAST analysis.

Overall, we detected a small number of virulence genes in the bioinformatic comparison with publicly available genome sequences. This is probably due to the fact that available databases report a restricted number of robustly annotated genes that are not specific for *Klebsiella* species, probably leading to an underestimation of their presence. For this reason, *K. michiganensis* 23999A2 was further analyzed with additional tools, and some virulence genes were identified (Table S2). In particular, the genes of the type I, III, and VI secretory pathways and the

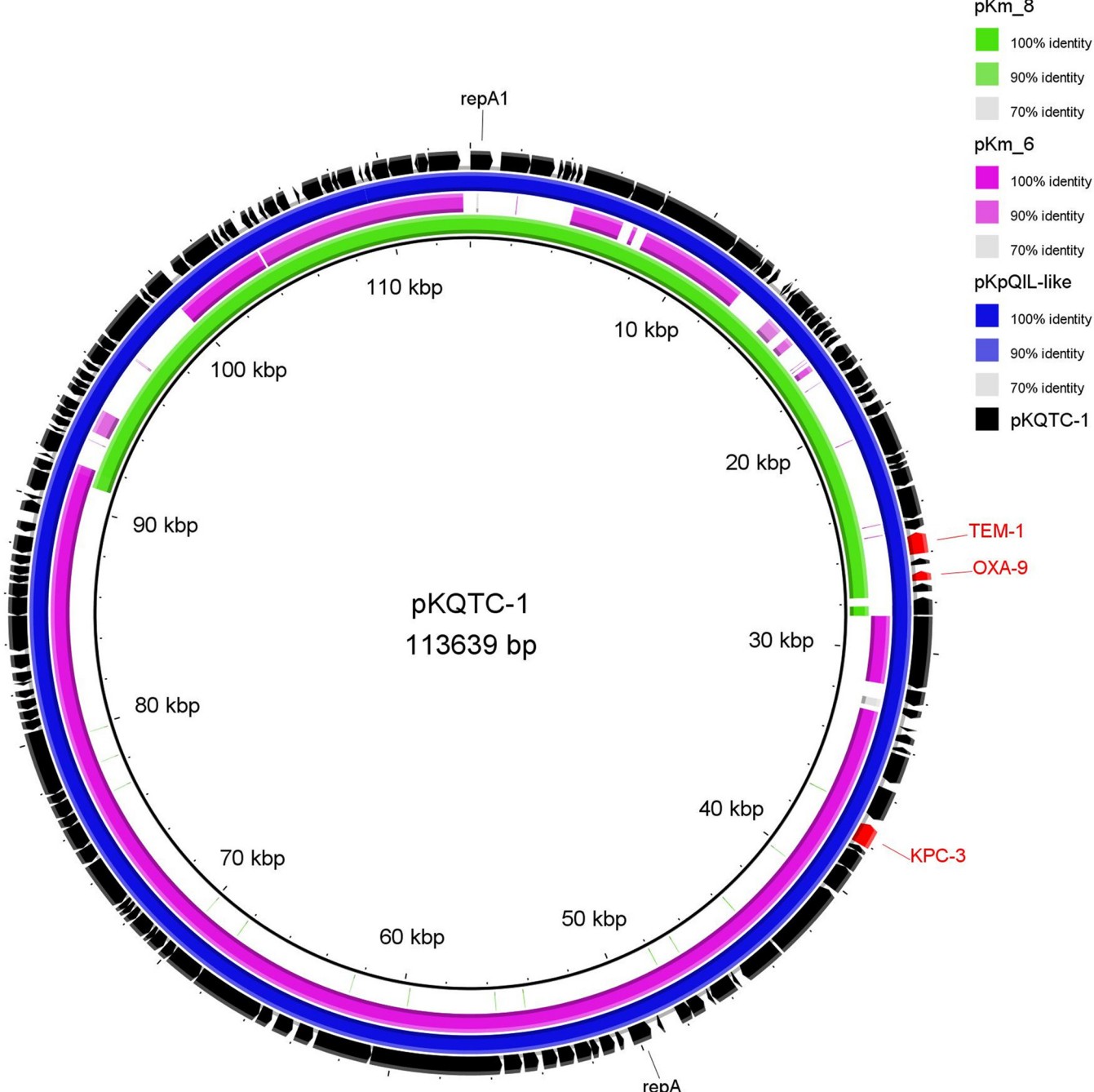

**FIG 3** Alignment of the pKQTC-1 plasmid of the *K. quasipneumoniae* tc-1 transconjugant with: the pKpQIL (pKPC) of the *K. pneumoniae* D1 and the pKm_6 and pKm_8 plasmids of the *K. michiganensis* 23999A2 donor strain. The BRIG software package, version 0.95, was used to create this image.

loci encoding aerobactin, salmochelin, and enterobactin siderophores were found. The *fim* and *mrk* loci encoding type 1 and type 3 fimbriae are involved in the intestinal colonization and biofilm formation of *K. pneumoniae*, which contribute to its pathogenicity. The three siderophore systems have been associated with infections in human clinical isolates of *K. pneumoniae* (1). Interestingly, all of these characteristics were also described in a *K. michiganensis* strain that was isolated from urban hospital effluents in South Africa (10).

The typing of the *Klebsiella* K and O loci, which encode the polysaccharide capsule and the lipopolysaccharide O antigen, was performed using the Kaptive tool (http://kaptive.holtlab .net/). However, the *K. michiganensis* 23999A2 was untypeable, due to its BLASTn coverage of <90% to the reference sequences and it missing more than three genes in the two

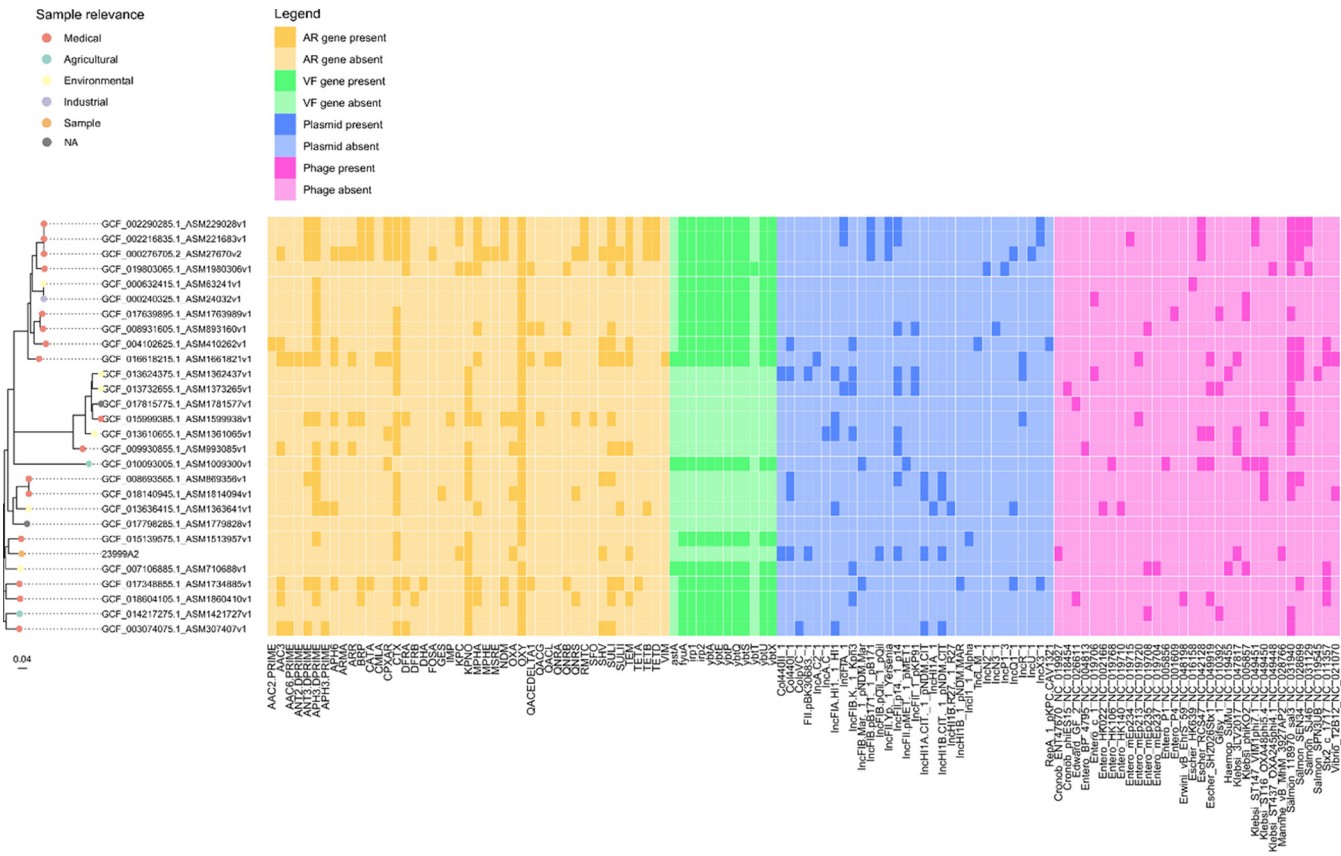

**FIG 4** Results of the antimicrobial resistance genes, virulence genes, plasmids, and phages analyses. The phylogenetic tree, based on the core-genome alignment, is reported on the left. In the "Sample relevance" legend, "Sample" indicates *K. michiganensis* 23999A2. AR, antimicrobial resistance; VF, virulence factor.

regions. In our strain, the allele types of the *wzi* and *wzc* genes that were included in the *cps* locus were identified as *wzi*-163 and *wzc*-917, respectively, and the best locus match was for capsule type KL107 (79% coverage and 77% identity). The O locus showed the highest similarity with the antigen O1/O2v1 (94% coverage and 72% identity). Although there is not a rigorous association of capsular type with ST or specific clones, KL107 has also been recently reported in the ST258 epidemic *K. pneumoniae* clone (21).

A pan-genome analysis (Fig. 5) performed on the *K. michiganensis* genomes showed that only 35% of the genes (3,271 out of 9,270) belong to the core genome. This result, together with the steady increase of the pan-genome size and the continuous addition of new genes accompanying each new genome sequence, suggest that *K. michiganensis* has an a open pan-genome, similar to that of *K. pneumoniae* (22). This finding agrees with the ability to adapt to different environmental conditions, which is typical of free-living microorganisms, such as *Klebsiella* spp. (23). Moreover, we observed the presence of a large accessory genome, which hints to a great potential for genomic plasticity for this species. In particular, genomic plasticity might play an important role in the adaptation of *K. michiganensis* to new environments and highlights the risk of the emergence of multidrug-resistant or hypervirulent clones.

In conclusion, this study confirms that carbapenem resistance genes are now disseminated in different *Enterobacteriaceae* and environments, including seafood, and also highlights that *K. michiganensis* species, similarly to *K. pneumoniae* (1) and thanks to its genome plasticity and plasmid permissive trait, could have a key role in the flow of new combinations of resistance and virulence genes from environmental sources to human pathogen populations.

## MATERIALS AND METHODS

**Bacterial strain isolation and identification.** The strain *K. michiganensis* 23999A2 was recovered from a clam sample during a study (Cariverona Project ID no. 9210) in which different environmental, human,

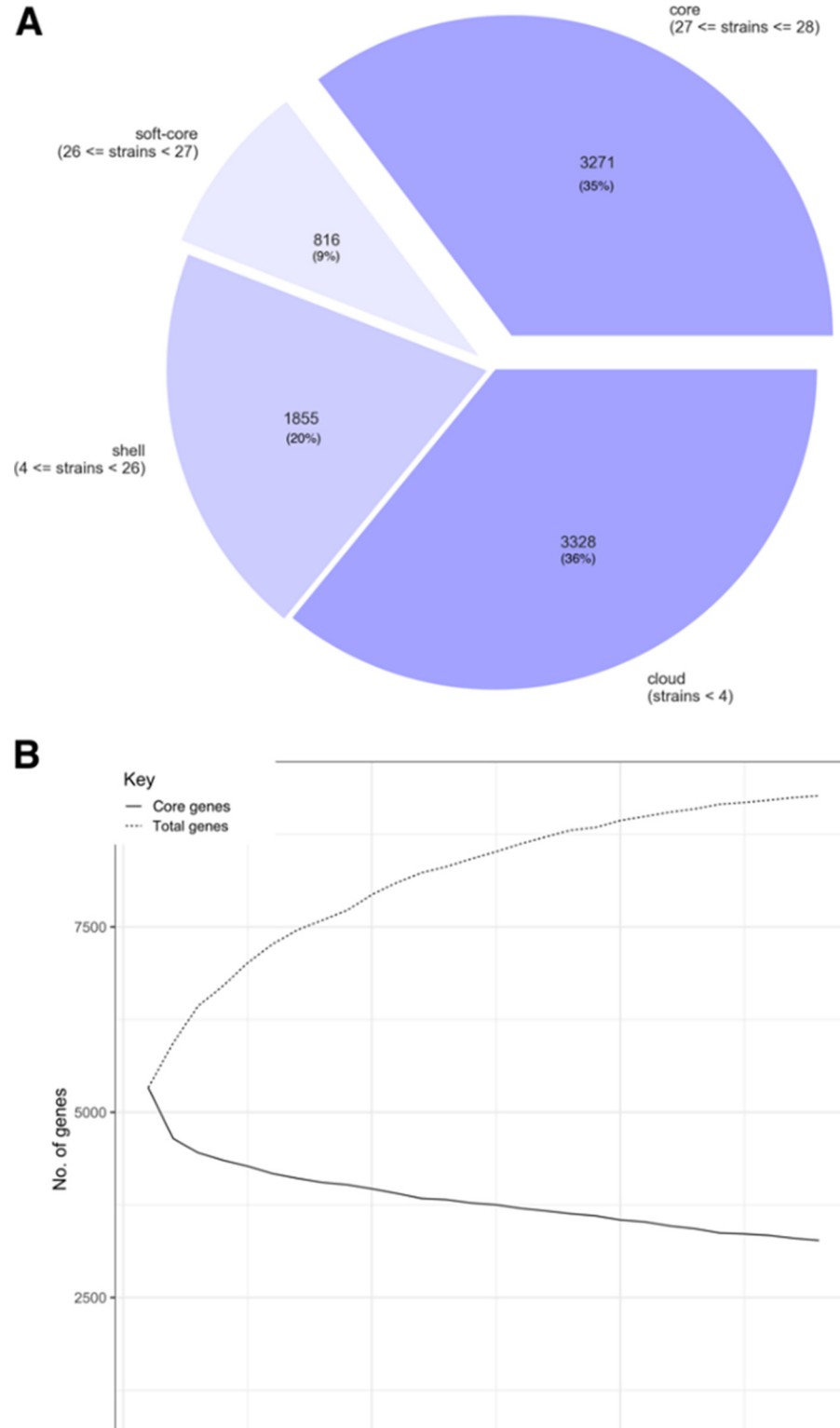

**FIG 5** Pan-genome analysis results. (A) Pie chart of the distribution of genes in the core, soft-core, shell, and cloud genes classifications. In total, 28 strains were analyzed. (B) Graph of the number of core and total genes when each analyzed genome was added to the pan-genome calculation.

and animal samples were screened for the presence of carbapenem-resistant strains. *Venus gallina* clam samples (*n* = 220) that were collected between November of 2018 and December of 2019 from different sampling sites at harvesting areas along the coast of the Adriatic Sea in the Marche region were analyzed. From each sample, approximately 50 g of shellfish flesh and intravalvular liquid were homogenized in buffered peptone water (1:10). 50 mL of the suspensions were diluted (1:1) in double strength minerals modified glutamate broth (MMGB, Liofilchem, Roseto degli Abruzzi, Italy) and incubated for 24 h at 37°C. Then, an enrichment step of 1:100 dilution in Luria-Bertani (LB) broth (Invitrogen, Merck KGaA, Darmstadt, Germany) supplemented with 0.12 $\mu$g/mL ertapenem (Sigma-Aldrich, Merck KGaA) was performed. After 24 h of incubation at 37°C, aliquots (100 $\mu$L) from the enrichment broth were subcultured in MacConkey agar plates supplemented with ertapenem (0.12 $\mu$g/mL; Merck Life Science). Presumptive ertapenem-resistant colonies were selected and isolated via streaking on the same growth medium. The bacterial identification of isolates was performed via MALDI-TOF (Bruker Daltonics, Bremen, Germany). The isolates were screened for carbapenemase production via the modified Hodge test (24).

**Antimicrobial susceptibility and PCR assays.** The susceptibility to imipenem, ertapenem, and meropenem was assessed via MIC determination, according to the procedures and breakpoints reported by EUCAST (version 12.0, www.eucast.org). *E. coli* ATCC 25922 was used for quality control. The detection of $bla_{KPC}$, $bla_{VIM}$, $bla_{NDM}$, and $bla_{OXA-48}$ was performed via multiplex PCR, using the primers and conditions reported by Poirel et al. (25).

**Mating experiments and plasmid curing.** The transferability of carbapenem resistance was investigated via conjugation experiments as previously described (26), using two different strains as recipients: the rifampin-resistant and nalidixic acid-resistant *E. coli* 1816 and the colistin-resistant *K. quasipneumoniae* ATCC700603 derivative. The resistance to colistin in *K. quasipneumoniae* ATCC700603 was conferred by the *in vitro* acquisition of a 33 kb IncX4 plasmid carrying *mcr-1.2* (27). This recipient strain also carried two plasmids of 152 kb and 77 kb in size, belonging to the IncFII$_K$ and IncM groups, respectively (28), with the 77 kb plasmid encoding the SHV-18 and OXA-2 ESBL (Table 1). Transconjugants were selected on brain heart infusion agar (BHIA) (Oxoid, Basingstoke, United Kingdom) containing ertapenem (1 $\mu$g/mL), rifampin (100 $\mu$g/mL), and nalidixic acid (100 $\mu$g/mL) or colistin (2 $\mu$g/mL), depending on whether *E. coli* 1816 or *K. quasipneumoniae* was used as the recipient strain. The transfer frequency was expressed as the ratio of the cell number (CFU/mL) of the transconjugant to that of the recipient. The transconjugants were evaluated for their susceptibility to carbapenems and were tested via PCR for the presence of *bla* genes.

In the curing assays, the strain 23999A2 was grown overnight in MacConkey agar (Oxoid) at 37°C for several passages (up to 40). Each week, 10 colonies were picked up, and their DNA was extracted and screened for the presence of the $bla_{KPC}$, $bla_{TEM}$, $bla_{SHV}$, and $bla_{OXA}$ genes via PCR.

The selected cured isolates and transconjugants were analyzed via WGS to confirm plasmid curing or acquisition, respectively.

**WGS and sequence typing.** WGS was performed using MicrobesNG Service (https://microbesng.com/) with the Illumina Miseq short read technology (2 × 250 paired-end) in combination with the Oxford Nanopore long reads. The assembled and annotated genome provided by the MicrobesNG Service was further analyzed using different tools (MLST v.2.0, ResFinder v.3.2, VirulenceFinder 2.0, and PlasmidFinder 2.1) that were available at the Center for Genomic Epidemiology (https://www.genomicepidemiology.org/) for resistance gene identification, strain typing, and plasmid profile determination. Virulence genes were also identified using VFanalyzer on the VFDB database (https://www.mgc.ac.cn/cgi-bin/VFs/v5/main.cgi?func=VFanalyzer). The sequence type of the strain was determined from WGS data, searching allelic matches on the PubMLST database of the *K. oxytoca* species complex (https://pubmlst.org/organisms/klebsiella-oxytoca). The OrthoANI analysis tool (https://www.ezbiocloud.net/tools/ani) and Ribosomal Multilocus Sequence Typing (rMLST) (https://pubmlst.org/species-id) were used as further species identification tools.

**Comparison with publicly available genome sequences.** A comparison with publicly available genome sequences was performed in order to investigate whether the *K. michiganensis* clam strain showed similarities with environmental or clinical strains as well as to identify possible geographical correlations. All of the available *K. michiganensis* complete genome sequences (*n* = 31) were downloaded from NCBI (https://www.ncbi.nlm.nih.gov, accessed on 04-11-2021; see Table S1 for details), and a taxonomic check was performed by calculating the average nucleotide identity (ANI) using the FastANI v1.33 tool with its default parameters (29). Genome sequences were considered to belong to the same species when the ANI value was ≥95%, whereas sequences having an ANI value of <95% with all available genomes were excluded from the following analyses, as they were possibly misclassified. Genomes were considered to be clinical when the NCBI "Sample relevance" field was "Medical", and genomes were considered to be environmental, otherwise (i.e., "Sample relevance" of "Agricultural", "Environmental", or "Industrial").

The core-genome SNP (single nucleotide polymorphism) tree in Newick format was generated by performing a core-genome alignment with the parsnp v1.2 tool of the Harvest-OSX64-v1.1.2 suite, using the -c parameter and specifying the sequenced *K.michiganensis* 23999A2 as the reference (-r parameter) (30). The tree file in Newick format was used as the input in the Microreact online tool (available at http://microreact.org), together with a metadata file (Table S1), for visualization.

The presence of antimicrobial resistance genes, virulence genes, and plasmids were assessed using Abricate v1.0.1 (https://github.com/tseemann/abricate) to search against the Megares, VFDB, and PlasmidFinder databases, respectively (accessed November 2021; *n* = 6,635, 2,597 and 460 nucleotide sequences, respectively). Sequences were considered present when covering at least 50% of the database sequence and showing 85% nucleotide identity. The presence of phages was also investigated using the Phage Search Tool Enhanced Release (PHASTER) online tool with its default parameters (31). Only phages classified as "intact" by the tool were considered to be present. Finally, the pan-genome was explored using the Roary v 3.13.0 tool (http://sanger-pathogens

.github.io/Roary), specifying the -e and -n parameters. Fig. 5 was generated using the roary_plots.py script (available at: https://github.com/sanger-pathogens/Roary/tree/master/contrib/roary_plots).

**Data availability.** The WGS data of the *K. michiganensis* 23999A2, of the cured strain, and of the transconjugant *K. quasipneumoniae* tc-1 are available under the BioProject ID PRJNA850894 (accession numbers: JANFNZ000000000, JANXJB000000000, and JANXJA000000000, respectively).

## SUPPLEMENTAL MATERIAL

Supplemental material is available online only.
**SUPPLEMENTAL FILE 1**, XLSX file, 0.03 MB.

## ACKNOWLEDGMENTS

This work was supported by "Fondazione Cariverona," Project ID no. 9210, "Detection of environmental reservoirs of carbapenem resistance", 2017, as well as by PRIN 2017 grant prot. 201728ZA49 from MIUR Italy. The funders had no role in the study.

F.L., E.R., and F.B. contributed to the collection and analysis of the clam samples as well as to the isolation of the strain. S.S. performed the susceptibility tests as well as the curing and conjugation experiments. M.C. and M.M.L. contributed to the methodology and the supervision of the molecular analysis. L.V. and G.M. performed the bioinformatic analysis. A.B., G.M., and E.G. contributed to the genome data analysis as well as the writing and review of the manuscript. C.V. was the coordinator of the project and contributed to the supervision of the study as well as the writing of the original draft. All of the authors were involved in the review of the final manuscript. No author has a conflict of interest to declare.

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
