## [Reviewer comments · Microbiology Spectrum]

Microbiology Spectrum

The emerging nosocomial pathogen *Klebsiella michiganensis*: genetic analysis of a KPC-3 producing strain isolated from Venus clam

Serena Simoni, FRANCESCA LEONI, Laura Veschetti, Giovanni Malerba, Maria Carelli, Maria Lleò, Andrea Brenciani, Gianluca Morroni, Eleonora Giovanetti, ELENA ROCCHEGIANI, Francesca Barchiesi, and Carla Vignaroli

Corresponding Author(s): Carla Vignaroli, Universita Politecnica delle Marche

Review Timeline:

Submission Date:	October 17, 2022
Editorial Decision:	November 23, 2022
Revision Received:	November 28, 2022
Accepted:	November 29, 2022

Editor: Sandeep Tamber

Reviewer(s): The reviewers have opted to remain anonymous.

Transaction Report:

DOI: <https://doi.org/10.1128/spectrum.04235-22>

November 23, 2022

Dr. Carla Vignaroli
Universita Politecnica delle Marche
Department of Life and Environmental Sciences
via Breccie Bianche
Ancona 60131
Italy

Re: Spectrum04235-22 (The emerging nosocomial pathogen *Klebsiella michiganensis*: genetic analysis of a KPC-3 producing strain isolated from Venus clam)

Dear Dr. Carla Vignaroli:

Link Not Available

Sincerely,

Sandeep Tamber

Journals Department
Editor comments:

The sequence analysis portion of the method needs more details. Please provide the parameters used for the bioinformatic programs, and specify the type of tree constructed.

Reviewer comments:

Reviewer #1 (Comments for the Author):

Your work is about genome analysis of a *K. michiganensis* strain, including PCR assays detected resistance genes, WGS, and analysis of plasmid and VFs. However, your article conclusion extend to the whole environment. Maybe your work did not provide sufficient evidence to prove your conclusion.

Reviewer #2 (Comments for the Author):

The authors have presented a relevant and interesting area of research which is currently lacking in the literature. The methods used were appropriate and robust with experimental data alongside bioinformatic review to confirm findings. The experimental data and the bioinformatics support one another. The data presented will further add to our understanding of genome plasticity of *Klebsiella* sp. and the requirement for surveillance of environmental isolates that have the potential to reach the human food chain to understand transfer of resistance between species and the emergence of multidrug resistant strains.

Suggested edits.

Line 75. In this study, a KPC-producing strain of *K. michiganensis* isolated from clams collected from natural beds along the Adriatic Sea coast in central Italy was were characterized by WGS.

76 natural beds along the Adriatic Sea coast in central Italy was characterized by WGS.

Line 86. MIC determination for ertapenem, imipenem and meropenem showed that the isolate was resistant to the three carbapenems with MIC values of 256, 16 and 8 µg/ml respectively.

Line 133. These findings also highlighted that pKm_8 could be originated from genetic recombination Requires some editing for grammar.

Staff Comments:

Preparing Revision Guidelines

Please return the manuscript within 60 days; if you cannot complete the modification within this time period, please contact me. If you do not wish to modify the manuscript and prefer to submit it to another journal, please notify me of your decision immediately so that the manuscript may be formally withdrawn from consideration by Microbiology Spectrum.

The emerging nosocomial pathogen *Klebsiella michiganensis*: genetic analysis of a KPC-3 producing strain isolated from Venus clam.

The authors recovered and characterised an isolate of multidrug resistant KPC-3 producing *Klebsiella michiganensis* from a Venus clam mollusc collected from natural beds in the Adriatic Sea coast in central Italy. Using whole genome sequencing genetic elements carrying beta lactam resistance genes were analysed and the ability of the strain to transfer carbapenem resistance by conjugation was investigated. The isolate was identified as being in ST382 with seven plasmid replicons.

The authors have presented a relevant and interesting area of research which is currently lacking in the literature. The methods used were appropriate and robust with experimental data alongside bioinformatic review to confirm findings. The experimental data and the bioinformatics support one another. The data presented will further add to our understanding of genome plasticity of *Klebsiella* sp. and the requirement for surveillance of environmental isolates that have the potential to reach the human food chain to understand transfer of resistance between species and the emergence of multidrug resistant strains.

Suggested edits.

Line 75. In this study, a KPC-producing strain of *K. michiganensis* isolated from clams collected from natural beds along the Adriatic Sea coast in central Italy ~~was~~ **were** characterized by WGS.

76 natural beds along the Adriatic Sea coast in central Italy was characterized by WGS.

Line 86. MIC determination for ertapenem, imipenem and meropenem showed **that the** isolate was resistant to the three carbapenems with MIC values of 256, 16 and 8 µg/ml respectively.

Line 133. These findings also highlighted that pKm_8 could be originated from genetic recombination **Requires some editing for grammar.**

"Response to Reviewers"

Editor comments:

The sequence analysis portion of the method needs more details. Please provide the parameters used for the bioinformatic programs, and specify the type of tree constructed.

As suggested further details on bioinformatic analysis and the type of phylogenetic tree have been provided.

Reviewer comments:

Reviewer #1 (Comments for the Author):

Your work is about genome analysis of a *K. michiganensis* strain, including PCR assays detected resistance genes, WGS, and analysis of plasmid and VFs. However, your article conclusion extend to the whole environment. Maybe your work did not provide sufficient evidence to prove your conclusion.

Our conclusions have been suggested not only by WGS analysis of our strain but also from pan-genome analysis of all *K. michiganensis* genomes included in the database.

Reviewer #2 (Comments for the Author):

The authors have presented a relevant and interesting area of research which is currently lacking in the literature. The methods used were appropriate and robust with experimental data alongside bioinformatic review to confirm findings. The experimental data and the bioinformatics support one another. The data presented will further add to our understanding of genome plasticity of *Klebsiella* sp. and the requirement for surveillance of environmental isolates that have the potential to reach the human food chain to understand transfer of resistance between species and the emergence of multidrug resistant strains.

Thank you to the reviewer for the positive comments.

Suggested edits.

Line 75. In this study, a KPC-producing strain of *K. michiganensis* isolated from clams collected from natural beds along the Adriatic Sea coast in central Italy was **were** characterized by WGS.

76 natural beds along the Adriatic Sea coast in central Italy was characterized by WGS.

The sentence was rephrased.

Line 86. MIC determination for ertapenem, imipenem and meropenem showed **that the** isolate was resistant to the three carbapenems with MIC values of 256, 16 and 8 µg/ml respectively.

As suggested the sentence has been corrected.

Line 133. These findings also highlighted that pKm_8 could be originated from genetic recombination **Requires some editing for grammar.**

The sentence has been modified.

November 29, 2022

Dr. Carla Vignaroli
Universita Politecnica delle Marche
Department of Life and Environmental Sciences
via Brecce Bianche
Ancona 60131
Italy

Re: Spectrum04235-22R1 (The emerging nosocomial pathogen *Klebsiella michiganensis*: genetic analysis of a KPC-3 producing strain isolated from Venus clam)

Dear Dr. Carla Vignaroli:

Your manuscript has been accepted, and I am forwarding it to the ASM Journals Department for publication. You will be notified when your proofs are ready to be viewed.

Sincerely,

Sandeep Tamber
Editor, Microbiology Spectrum
